

# Evaluating field-goal shooting effectiveness in wheelchair basketball players across a competitive season: a preliminary study

Valentina Cavedon[1], Marta Zecchini[1], Marco Sandri[2], Paola Zuccolotto[2], Caterina Biasiolo[1], Carlo Zancanaro[1] and Chiara Milanese[1]

[1] Department of Neurosciences, Biomedicine and Movement Sciences, University of Verona, Verona, Italy
[2] BODaI-Lab, University of Brescia, Brescia, Italy

## ABSTRACT

**Background**. Information about non-elite wheelchair basketball (WB) players across national competitive seasons are still missing. This study aimed at identifying which situational-related variables were associated with shooting effectiveness in non-elite WB players.

**Methods**. All the matches played by one WB team across one national competitive season were video-recorded and analysed; 333 shooting attempts from high-point players and several situational-related variables were considered.

**Results**. Pearson's Chi-square test showed that increased shooting effectiveness under the following conditions: playing on home ground, during won matches, while taking shots with the wheelchair in motion, and when no opposing player raised their arm in defence. Results of the multivariable logistic regression analysis showed a statistically significant influence of match location ($p$-value $= 0.001$), shot-clock remaining ($p$-value $= 0.015$) and modality of press ($p$-value $< 0.001$). The highest attack effectiveness was achieved when teams played at home (odds ratio [OR] $= 2.49$), while the shooting effectiveness decreased when the shot occurred during the last seconds of the action (OR $= 0.36$), or the opponents defended with the arm raised (OR $= 0.19$). These results suggest that coaches should include exercises aimed at shooting under conditions of increased pressure in their programmes in order to create specific situations during the training sessions to prepare their high-point athletes for shots under specific match constraints.

Corresponding author
Valentina Cavedon,
valentina.cavedon@univr.it

# INTRODUCTION

Wheelchair basketball (WB) is a popular and inclusive adapted team sport practiced in nearly 100 countries worldwide (*Cavedon, Zancanaro & Milanese, 2018*; *Arroyo et al., 2022*). Around the world, opportunities for both males and females to participate in WB are increasing rapidly at international and national levels, as well as at recreational, collegiate, and junior levels.
In WB, the regulations are very similar to those of running basketball (hereinafter "basketball"), like the court size with identical basket heights, the scoring systems, and the number of players per team on the court at any one time. Due to the presence of players with different types and degrees of severity of their disabilities (*e.g.*, spinal cord injury, lower limb amputation/s, joint and musculoskeletal conditions), WB players are classified using a scoring system ranging from 1.0 point to 4.5 points on an ordinal scale (with 0.5-point increments) according to their ability to carry out the activities required by the game (*i.e.,* move the chair, pivot, shoot to basket, bounce, and so on). The lowest score (1.0 point) is assigned to players with minimal functional potential, like players with a complete spinal cord injury at the thoracic level, while the highest score (4.5 points) is assigned to players with maximal functional potential, like players with unilateral lower limb amputation below the knee (*International Wheelchair Basketball Federation, 2021*). To ensure balanced competitions, the sum of scores for the five players on the court at any time cannot exceed 14 points (*International Wheelchair Basketball Federation, 2021*).

Wheelchair basketball requires the integration of sport-specific skills (*i.e.,* shooting, dribbling, and passing) and frequent bouts of high-intensity movements in complex technical-tactical scenarios (*Bloxham et al., 2001*; *Wang et al., 2005*; *Gil et al., 2015*). As in basketball, among the various sport-specific skills, in WB shooting is the way to obtain points and, therefore, the player's skill in throwing the ball through the rim has been identified as a key indicator of success for both the team and individual athlete (*Malone, Gervais & Steadward, 2002*; *Oudejans et al., 2012*; *Francis, Owen & Peters, 2021*; *Alsasua et al., 2021*). However, in WB this skill is even more difficult than in basketball. The rim of the basket is the same height as in regular basketball, but WB players sit in a low position and, in addition, they are unable to use upward leg force to help project the basketball (*Malone, Gervais & Steadward, 2002*; *Goosey-Tolfrey, Butterworth & Morriss, 2002*; *Goosey-Tolfrey, Butterworth & Morriss, 2002*). During elite WB games, it has been shown (*Zwakhoven et al., 2003*; *Francis, Owen & Peters, 2021*) that one of the most important technical skills required by players is field-goal shooting *i.e.,* any legal shot attempted and/or made by a WB player in a game during live action. In fact, during elite WB games most shots occurred during live actions, with an average of between 57 and 64 field-goal shots per game, compared with an average of between 11 and 16 free throw attempts per game (*Gómez et al., 2014*; *Gómez et al., 2015*; *Francis, Owen & Peters, 2021*).

From a tactical modelling perspective, the field-goal shoot in WB is a complex skill that depends on several situational-related variables, like game status, shot clock remaining, shot location (*Francis, Owen & Peters, 2019a*; *Francis, Owen & Peters, 2021*). For example, it has been shown (*Francis, Owen & Peters, 2021*) that field-goal attempts taken earlier in the possession resulted in a significantly higher rate of success and that the classification of the player also has an impact on field-goal shooting effectiveness. Concerning the classification of the players, researchers (*Gómez et al., 2014*; *Francis, Owen & Peters, 2021*) acknowledged that high-point players (*i.e.,* players with fewer movement restrictions and classified from 3.5–4.5 points) are those who typically attempt more shots during a game with a higher shooting effectiveness, in addition to being those with a more instrumental role in the game (*Vanlandewijck et al., 2004*; *Skučas et al., 2009*). Accordingly, a recent study

by *Francis, Owen & Peters (2021)* indicated that WB teams should consider prioritizing the opportunities for high-point players to attempt the field-goals, regardless of other situational-related variables (*i.e.,* the time left on the clock or the location on the court).

Performance analysis is one of the newest sport science disciplines, which involves the labelling and recording of sport specific actions and behaviours, with the integration of coaches' knowledge (*Sampaio, 2013*). While the use of data science for a wide variety of purposes is well established in basketball (*Zuccolotto & Manisera, 2020*), a lower number of contributions can be found in the literature regarding WB, especially with reference to non-elite teams and players. This is mainly due to the unavailability of the large and complete datasets that are instead often guaranteed in basketball. More specifically, performance analysis represents an interesting tool for a better understanding and interpretation of WB games in terms of technical requirements and tactical responses thereby improving the training sessions and better preparing the players for competition according to real game constraints (*Gómez et al., 2014*). To date, most of the available literature dealing with performance analysis in WB refer to elite players (*De Witte et al., 2017*; *Francis, Owen & Peters, 2019b*; *Francis, Owen & Peters, 2021*) and scientific evidence derived from data collected during high-level tournaments like the Paralympic Games and the World Championships (*Gómez et al., 2014*; *Gómez et al., 2015*; *Francis, Owen & Peters, 2019a*). However, information about non-elite WB players across national competitive seasons are still missing and therefore coaches and operators in this field can only rely on their personal experience and/or on data derived from high-level tournaments. It is reasonable to assume that non-elite tournaments reserved to club teams would have specific characteristics which differ from that of elite tournaments in which national teams participate. For example, national tournaments differ by the Paralympic games or well the World Championships in the fact that national tournaments take place over several months and matches are played either at home (*i.e.,* usually in the gym where athletes train) or away from home (*i.e.,* in a gym where there may be a lot more crowd encouragement for the opponents and that may require a long journey to reach). Accordingly, there is the need to provide evidence-based information about non-elite WB games played across a competitive season in order to identify which performance parameters are specific for this type of tournament as well as at this competitive level.

As a first step toward this aim, we conducted a video analysis of the matches played by a non-elite WB team over an entire competitive season, focusing on all attempted field-goal shots. According to the literature (*Gómez et al., 2014*; *De Witte et al., 2016*; *Molik et al., 2017*; *Francis, Owen & Peters, 2021*; *Arroyo et al., 2022*), given the impact of the functional class in WB shooting effectiveness as well as in shooting strategy and considering that high-point players are those who have more chances to attempt field-goals during the games, in this paper we decided to consider exclusively the offensive sequences ending in a shot by a high-point player with a twofold aim. First, to identify the situational-related variables (*e.g.,* the time and location of the match, the partial and final result of the match and the zone from which the player attempted the shot) which are associated with shooting effectiveness and, second, to explore the impact of each situational-related variable upon the effectiveness of shooting using a multivariate logistic regression. Providing evidence about

game performance for non-elite WB players based on their functional class would assist coaches in tailoring and adapting training sessions to better reflect the actual demands of the game for each player. This information would also support a coach' decisions regarding the specific requirements of the competitive level in which their athletes participate.

## MATERIALS & METHODS

### Sample

In line with the aims of our study, we applied a non-participative observational approach. The sample consisted of all the shots attempted by one WB team during the matches ($n = 8$) from the regular 2021/2022 competitive season of the Italian Wheelchair Basketball Championship ("Campionato di Serie B—Trofeo Antonio Maglio"). Members of the evaluated team ($n = 5$) were all males and they had been playing WB at a competitive level for at least 10 years. Disability comprised transfemoral amputation ($n = 2$), transtibial amputation ($n = 1$), polio ($n = 1$) or congenital neurological disorder ($n = 1$). Players trained regularly for 2 training sessions per week of 2 h each.

The considered tournament is managed by the Federazione Italiana Pallacanestro in Carrozzina (Italian Wheelchair Basketball Federation, FIPIC) (*Federazione Italiana Pallacanestro in Carrozzina, 2021*). The technical and medical regulations governing the championship are set out by the FIPIC under the surveillance of the Italian Paralympic Committee, the International Paralympic Committee, and the International Wheelchair Basketball Federation. All the procedures performed in the study were in accordance with the Declaration of Helsinki as well as with the ethical standards of the local ethics committee, and the protocol was approved by the Institutional Review Board of the local University (Protocol number: 18198, 05/04/2013). The University of Verona granted Ethical approval to carry out the study. All the participants were informed about the aims of the study and the experimental procedures, and they knew that they could withdraw at any time. All participants read and signed the informed consent form.

### Data collection and handling procedures

The obtained game footage was filmed from an elevated position at the half-way line and provided a half-court perspective with an overlay of the time clock and current scoreboard. All data were annotated by an experienced observer graduated in Sports Sciences, with a technical coaching license issued by the FIPIC and with great experience as WB analyst in the considered league. The operator spent about 2 h and 30 minutes analysing each game. On any given day, a maximum of two games were analysed in an attempt to reduce errors and a five-minute break was taken at the end of each quarter (*Liu, Jaramillo & Vincenzi, 2015*). All the attack situations were re-observed to test data reliability and no adjustments to the analysed data were necessary. To assess the reliability of data, another expert also evaluated independently the shooting actions.

One game (*i.e.*, the first match of the tournament) was not available due to technical problems in data acquisition and, therefore, the seven available games were analysed over a three-month period. A total of 577 attack situations were gathered from the analysed matches, of which 518 ended with a shot. All the free-shots and the field-goals attempted

by low-point players were excluded from analysis. Thus, 333 shooting attempts from high-point players were considered and were transformed into dichotomous dependent variables (*i.e.,* successful/unsuccessful). The independent variables were related to situational-related variables according to *Francis, Owen & Peters (2019a)*, with some modifications. The independent variables considered were:

- Match location: playing at home or away.
- Result of the match: winning or losing.
- Match status: winning, loosing or drawing conditions.
- Quarter: quarter 1, quarter 2, quarter 3 and quarter 4.
- Shot-clock remaining: 24–13 s, 12–7 s, 6–0 s.
- Shot location: zone 1 (2 point—left base, 2 point—right base, 2 point—left 45, 2 point—right 45), zone 2 (2 point—centre near), zone 3 (2 point—centre mid, 2 point—centre long), zone 4 (2 point—left elbow, 2 point—right elbow) (Fig. 1).
- Shot-position modality: stationary or moving shot.
- Defensive modality: modality of press (no opponent with a raised arm, one or more opponents with raised arms), and man press (no press, one man, two or more men).

## Statistical analysis

The reliability of data (intra-rater agreement) was determined by Krippendorff'alpha (*Hayes & Krippendorff, 2007*). The Pearson's Chi-square test was used to analyse the effects between the shooting outcome (successful/unsuccessful) and the situational-related variables. Effect sizes (ES) were calculated using Cramer's V test and interpreted as small (ES $\leq$ 0.10), medium (0.1 $<$ ES $\leq$ 0.3) and large (ES $=$ 0.50) (*Volker, 2006*). In the case of any statistically significant association between an independent variable with three or more categories and the dependent variable, a *post-hoc* analysis with Bonferroni's correction was conducted in order to assess the differences between pairs of subgroups.

A multivariable logistic regression model was used to assess the relationship between situational-related variables and the binary response variable (0 and 1 for unsuccessful and successful shots, respectively). Then, the multivariable logistic regression model can be expressed as follows:

$$P(Y = 1|\mathrm{X}) = \frac{1}{1+e^{-z}} = \frac{1}{1+e^{-\mathrm{X}\beta}},$$

where $P(Y = 1|X)$ indicates the probability to score a basket given the observed variables X and where $Z = X \cdot \beta$ indicates the linear part of the logistic model given from $Z = \beta_0 + \beta_1 \cdot \mathrm{ML} + \beta_2 \cdot \mathrm{RM} + \beta_3 \cdot \mathrm{MS} + \beta_4 \cdot \mathrm{MP} + \beta_5 \cdot \mathrm{SC} + \beta_6 \cdot \mathrm{SL} + \beta_7 \cdot \mathrm{SPm} + \beta_8 \cdot \mathrm{ModP} + \beta_9 \cdot \mathrm{ManP}$. In the above-reported model $\beta_0$ indicated the constant of the equation and $\beta_1, \beta_2, \ldots, \beta_9$ are the coefficients for the independent variables, respectively ML (match location), RM (result of the match), MS (match status), MQ (match quarter), SC (shot-clock remaining), SL (shot location), SPm (shot-position modality), ModP (modality of press), ManP (man press). The odds ratio (OR) with its 95% confidence intervals (CI) and the likelihood ratio test (LRT) were computed for each explanatory variable in the model.

Statistical significance was set at *p*-value $\leq$ 0.05. All statistical analysis was performed with SPSS version 16.0 (IBM Corp., Armonk, New York, USA).

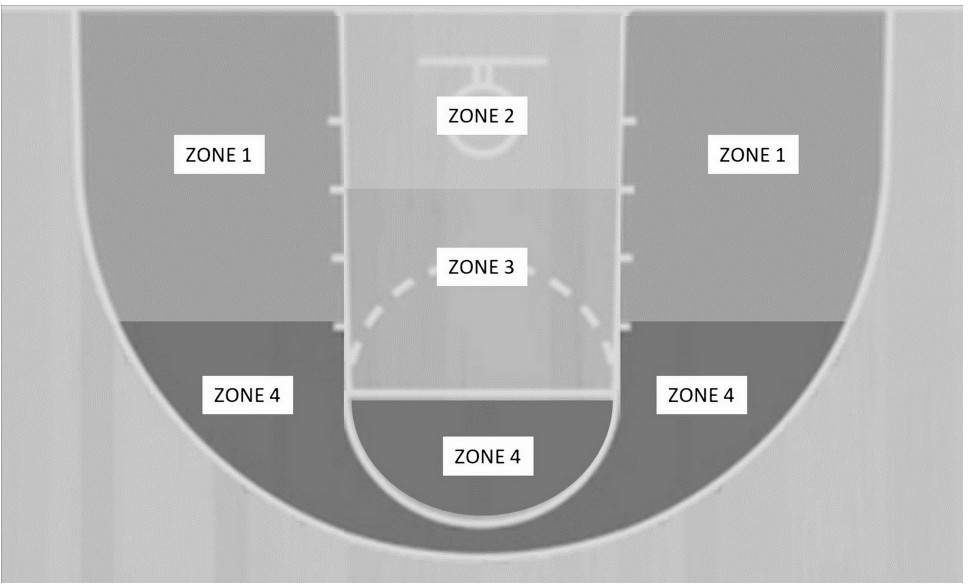

**Figure 1  Zones of the field.**

## RESULTS

The Krippendorff'alpha was greater than 0.95, thereby indicating a good agreement.

The sample distribution of each situational variable is summarized in Table 1 (percentage and case numbers). As shown in Table 1, no significant associations were found between shooting effectiveness and match quarter or man press, while all other investigated situational variables exhibited a statistically significant relationship with shooting effectiveness. In particular, the results indicated increased shooting effectiveness under the following conditions: playing on home ground, during won matches, while taking shots with the wheelchair in motion, and when no opposing player raised their arm in defence. Moreover, the results showed higher shooting effectiveness when the shot was attempted in a winning condition compared to a losing one ($p$-value $< 0.001$), and when the shot occurred during the first 24–13 s of the action, rather than in the last seconds of the action (*i.e.,* 6–0 s). Regarding the shooting zone, the *post-hoc* analysis did not reveal any statistically significant findings. All the results showed a medium ES.

Results of the multivariable logistic regression analysis (Table 2) showed a statistically significant influence of match location (LRT $= 10.34$, $p$-value $= 0.001$), shot-clock remaining (LRT $= 8.363$, $p$-value $= 0.015$) and modality of press (LRT $= 16.176$, $p$-value $< 0.001$). The highest attack effectiveness was achieved when teams played at home (OR $= 2.49$), while the shooting effectiveness decreased when the shot occurred during the last seconds of the action (OR $= 0.36$) or the opponents defended with the arm raised (OR $= 0.19$) (Table 2).

**Table 1  Frequency (%) and numerical (*n*) distribution of shooting effectiveness according to situational-related variables.**

| | Shooting effectiveness | | | | Chi-squared test | | |
|---|---|---|---|---|---|---|---|
| | Successful *n* = 140 | | Unsuccessful *N* = 193 | | | | |
| | % | *n* | % | *n* | $\chi^2$ | *p* | ES |
| Match location | | | | | | | |
| Playing at home | 52.3 | 78 | 47.7 | 71 | 11.756 | 0.001 | 0.188 (medium) |
| Playing away | 33.7 | 62 | 66.3 | 122 | | | |
| Match results | | | | | | | |
| Won match | 49.8 | 100 | 50.2 | 101 | 12.368 | <0.001 | 0.193 (medium) |
| Lost match | 30.3 | 40 | 69.7 | 92 | | | |
| Match status | | | | | | | |
| Winning condition | 53.4 | 79 | 46.6 | 69 | 14.636 | 0.001 | 0.210 (medium) |
| Loosing condition | 31.9 | 51 | 68.1 | 109 | | | |
| Drawing condition | 40.0 | 10 | 60.0 | 15 | | | |
| Match quarter | | | | | | | |
| Quarter 1 | 45.7 | 32 | 54.3 | 38 | 4.001 | 0.261 | 0.110 (medium) |
| Quarter 2 | 41.6 | 37 | 58.4 | 52 | | | |
| Quarter 3 | 47.7 | 42 | 52.3 | 46 | | | |
| Quarter 4 | 33.7 | 29 | 66.3 | 57 | | | |
| Shot-clock remaining | | | | | | | |
| 24–13 s | 48.7 | 76 | 51.3 | 80 | 7.625 | 0.022 | 0.151 (medium) |
| 12–7 s | 40.0 | 48 | 60.0 | 72 | | | |
| 6–0 s | 28.1 | 16 | 71.9 | 41 | | | |
| Shot location | | | | | | | |
| Zone 1 | 40.0 | 30 | 60.0 | 45 | 8.466 | 0.037 | 0.159 (medium) |
| Zone 2 | 49.7 | 78 | 50.3 | 79 | | | |
| Zone 3 | 32.9 | 23 | 67.1 | 47 | | | |
| Zone 4 | 29.0 | 9 | 71.0 | 22 | | | |
| Shot-position modality | | | | | | | |
| Stationary | 39.5 | 107 | 60.5 | 64 | 3.911 | 0.048 | 0.108 (medium) |
| In movement | 53.2 | 33 | 46.9 | 29 | | | |
| Modality of press | | | | | | | |
| Raised arm | 38.0 | 107 | 62.0 | 164 | 10.580 | 0.001 | 0.178 (medium) |
| No raised arm/s | 61.4 | 33 | 38.6 | 29 | | | |
| Man press | | | | | | | |
| No man press | 47.4 | 27 | 52,.6 | 30 | 1.820 | 0.403 | 0.074 (low) |
| 1 man press | 38.0 | 54 | 62.0 | 88 | | | |
| 2 or more men press | 44.0 | 59 | 56.0 | 75 | | | |

**Notes.**
$\chi^2$, Pearson's Chi-square; *p*, *p*-value; ES, effect size (Cramer's V).

# DISCUSSION

To the best of our knowledge, this is the first study investigating the field-goal effectiveness of non-elite high-point WB players across a competitive season. As the first aim, this

**Table 2** Results of multivariable logistic regression analysis relative to shooting effectiveness as a function of situational-related variables.

|  | OR | 95% CI | *p* |
|---|---|---|---|
| Match location |  |  |  |
| Playing at home | 2.49 | 1.42–4.38 | 0.001 |
| Match results |  |  |  |
| Won match | 1.40 | 0.62–3.15 | 0.419 |
| Match status |  |  |  |
| Winning condition | 1.30 | 0.46–3.69 | 0.618 |
| Losing condition | 1.03 | 0.32–3.37 | 0.960 |
| Match quarter |  |  |  |
| Quarter 1 | 1.98 | 0.90–4.36 | 0.089 |
| Quarter 2 | 1.56 | 0.79–3.06 | 0.201 |
| Quarter 3 | 1.81 | 0.93–3.55 | 0.083 |
| Shot-clock remaining |  |  |  |
| 6–0 s | 0.36 | 0.17–0.74 | 0.006 |
| 12–7 s | 0.86 | 0.50–1.49 | 0.598 |
| Shot location |  |  |  |
| Zone 1 | 1.37 | 0.52–3.58 | 0.524 |
| Zone 2 | 1.27 | 0.49–3.29 | 0.625 |
| Zone 3 | 0.73 | 0.27–2.02 | 0.550 |
| Shot-position modality |  |  |  |
| Stationary | 1.07 | 0.54–2.14 | 0.847 |
| Modality of press |  |  |  |
| Raised arm | 0.19 | 0.08–0.44 | <0.001 |
| Man press |  |  |  |
| No man press | 0.42 | 0.17–1.06 | 0.067 |
| 1 man press | 0.79 | 0.45–1.39 | 0.418 |

**Notes.**
OR, odds ratio; CI, confidence intervals; *p*, *p*-value.

study identified the situational-related variables (*e.g.*, the time and place of the match, the partial and final results of the match and the zone from which the player attempted the shot) associated with shooting effectiveness. The second aim of this study was to explore the impact of each situational-related variable upon the effectiveness of shooting using a multivariate logistic regression. Situational-related variables (see 'Materials and Methods' section) related to all the matches played by a non-elite WB team across one competitive season were identified through match-analysis and were selected according to the literature (*Francis, Owen & Peters, 2021*).

The initial finding of the present study was that the match location was associated with shooting effectiveness, with a ~18% higher shooting effectiveness when the team played at home. This result suggests that playing a league match in the same gym where training sessions are usually held increases the number of successful shots made in relation to the number of shots attempted. This result is in line with previous literature in team sports like football, volleyball and basketball showing that home advantage has an important role in

determining the outcome of a match (*Gómez et al., 2010*). For example, in basketball it has been found (*Pollard & Gomez, 2007*) that 60% of NBA matches are won by the home team and showed that the home advantage in European leagues was higher than in NBA, with percentages ranging from 63% (France) to 66% (Italy and Greece). The home advantage can be explained by the fact that when attempting a shot in an unknown environment there are several contextual-related variables to which the player is not familiar that may affect the shooting success, such as the position and/or intensity of the lights, travel, referee bias and crowd influence (a higher proportion of spectators favouring the home team) (*Pollard, 2008*). This result suggests that coaches should increase environmental variability in which their athletes train as much as possible by, for example, varying the intensity of the lights or by reproducing the crowd influence (*e.g.*, by exposing them to distracting sound through earbuds).

The results of the present findings highlighted that non-elite high-point WB players had a ~20% higher shooting effectiveness in the matches that were won in comparison to the matches lost. This underlines the importance of successful field-goal shooting in non-elite WB matches. This is in line with the literature dealing with basketball and elite WB showing that the field-goal shooting effectiveness is widely related to winning a match (*Sampaio & Janeira, 2003*; *Malarranha et al., 2013*; *Francis, Owen & Peters, 2021*). Moreover, the present results showed that the shooting effectiveness was higher than 50% when the player attempted the shot while their team was winning. This result is to be expected because there is a multiplicity of factors that determine the team success, including the fact that in some matches, when there was such a significant score difference in favour of a team, the opponents lose much of their defensive motivation. Moreover, the players of the winning team clearly experience reduced pressure for success which would seem to favour shot success.

As far as the match quarter is concerned, the results showed that the number of successful shots in relation to the number of shots attempted was almost similar in the 4 match quarters indicating that the match quarter did not affect the shooting effectiveness. These findings were similar to previous literature on elite WB (*Francis, Owen & Peters, 2021*) as well as on basketball (*Vaquera et al., 2016*; *Vencúrik et al., 2022*), where the match quarter did not show a significant impact on the shooting effectiveness. The maintenance of a constant shooting effectiveness throughout the four match quarters can be explained by the fact that the athletes examined may have been at a level of physical fitness adequate to the demands of the matches at their competitive level (*Vencúrik et al., 2022*). This finding of constant shooting efficacy may also be related to the fact that players in the first quarter, although not suffering from fatigue, may still need to adapt to new environmental factors. Later, in successive quarters, fatigue would be compensated by improved tactical goals and longer more controlled offensive plays leading to shots from better planned positions (*Scanlan et al., 2015*; *Vázquez-Guerrero et al., 2019*).

Another finding of this study was that field-goal attempts taken earlier in the possession resulted in significantly higher chances of success. This finding draws a parallel to elite WB as well as to basketball (*Skinner, 2012*) and is given by the fact that in the early stages of possession, the defence has less time to organise themselves and counter offensive

threats giving an unstable, unbalanced and disorganized early defence (*Lago-Ballesteros, Lago-Peñas & Rey, 2012*). A miss-match situation or a disorganized defensive team favours the offensive team to dictate play and to exploit their opportunities (*Francis, Owen & Peters, 2021*). In particular, it has been shown (*Remmert & Chau, 2019*) that, when the defensive team is unbalanced, the offensive team's high-point players tend to be countered by lower-point players of the defensive team. This allows offensive players to attempt their preferred shots early in a possession and, potentially, against less well-matched defenders (*Francis, Owen & Peters, 2021*). Then, in the subsequent stages of possession the balance favours the defensive team with offensive players having to take more rushed actions and through well-matched defenders (*Calvo, García & Navandar, 2017*). From a tactical perspective, this suggests that teams should focus on converting transitional offences into optimal shooting opportunities within the first seconds of possession (*Francis, Owen & Peters, 2021*).

Regarding shot location, the results of our findings revealed a statistically significant association between the zone from which the shot was attempted and its effectiveness. However, no statistically significant result was found in the *post-hoc* analysis. This may be due to the relatively small sample size which did not allow for more in-depth analyses. However, from the observation of the percentages of shooting effectiveness with regards to each zone, we notice that non-elite high-point WB players reach the highest percentage of shooting effectiveness (*i.e.,* ~50%) when attempting the shot from zone 2 (that corresponds to the 2-point-Centre-Near zone from the article by *Francis, Owen & Peters (2021)*). The percentage of shooting effectiveness was then reduced by about 10 percentage points when the shot was attempted from zone 1 (that corresponds to 2 point—left base, 2 point—right base, 2 point—left 45, 2 point—right 45 zones from the article by *Francis, Owen & Peters (2021)*). Finally, the lowest shooting effectiveness was observed when players attempted the shot in zones 3 (corresponding to 2 point—centre mid, 2 point—centre long zones from the article by *Francis, Owen & Peters (2021)*) and 4 (corresponding to 2 point—left elbow, 2 point—right elbow from the article by *Francis, Owen & Peters (2021)*), *i.e.,* further away from the basket. Therefore, it would seem that non-elite WB teams should look to prioritise the location closest to the basket to make a field-goal shot attempt, but this aspect requires further exploration in future studies with a larger number of field-goal attempts.

From a defensive action point of view, our results showed that the number of men pressing the player who made the shot did not affect shooting efficacy, whereas the modality of press did. More specifically, it has been shown that players had a higher probability of completing a shot when no opponent defended with their arm raised. In line with previous findings in basketball, these results showed that increasing the defensive pressure significantly reduces the percentage of success of shooting performance (*Alvarez et al., 2009*; *Gorman & Maloney, 2016*). Moreover, based on our results, WB players at a non-elite level showed reduced shooting effectiveness when playing away, when the shot was attempted during the lasts seconds of an action and with an opponent with their arm raised.

Taken together these results suggest coaches should include exercises aimed at shooting under conditions, of increased pressure in their training programmes, simulating the

presence of an opponent with an arm raised when attempting a shot. Coaches dealing with non-elite high-point WB players are encouraged to replicate the real match scenarios found at the relevant level of competition and to create specific situations during the training sessions to prepare their high-point athletes for shots under specific match constraints.

The present study has some limitations to take into account. First, the results presented herein refer to only one WB team. Accordingly, future research focusing on a larger number of non-elite WB teams is needed to further explore issues on match performance of athletes across a competitive season. Secondly, future research should also take into account further specific match-related variables (*e.g.*, pre-shot conditions or defensive pressure) which would lead to a better understanding of field-goal shooting effectiveness in a non-elite high-point WB players.

## CONCLUSIONS

In conclusion, the results of the present study delve into the specificity of the WB match at a on-elite competitive level. After having detected similarities and differences with elite WB matches, data is provided regarding the preparation of training programmes and the handling of competitions by coaches and technicians considering the specific match constraints of a non-elite competitive level. Moreover, the results of the present work allowed us to provide an initial characterization of the construction of offensive actions that would lead to more successful shots from the high-point players.

## ACKNOWLEDGEMENTS

The authors would like to thank all players and their technical staff, Olympic Basket Verona, and Federazione Italiana Pallacanestro in Carrozzina (FIPIC) for their kind support.

### Funding
The authors received MNO funding for this work. The funders had no role in study design, data collection and analysis, decision to publish, or preparation of the manuscript.

### Grant Disclosures
The following grant information was disclosed by the authors:
MNO funding for this work.

### Competing Interests
The authors declare there are no competing interests.

### Author Contributions
- Valentina Cavedon conceived and designed the experiments, analyzed the data, prepared figures and/or tables, authored or reviewed drafts of the article, and approved the final draft.
- Marta Zecchini conceived and designed the experiments, performed the experiments, prepared figures and/or tables, authored or reviewed drafts of the article, and approved the final draft.
- Marco Sandri analyzed the data, authored or reviewed drafts of the article, and approved the final draft.
- Paola Zuccolotto analyzed the data, authored or reviewed drafts of the article, and approved the final draft.
- Caterina Biasiolo performed the experiments, prepared figures and/or tables, authored or reviewed drafts of the article, and approved the final draft.
- Carlo Zancanaro analyzed the data, authored or reviewed drafts of the article, and approved the final draft.
- Chiara Milanese conceived and designed the experiments, authored or reviewed drafts of the article, and approved the final draft.

## Human Ethics

The following information was supplied relating to ethical approvals (*i.e.*, approving body and any reference numbers):

The University of Verona granted Ethical approval to carry out the study.

## Data Availability

The raw measurements are available in the Supplementary Files.

## Supplemental Information

Supplemental information for this article can be found online at http://dx.doi.org/10.7717/peerj.15785#supplemental-information.

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
