# Peer review of "Evaluating field-goal shooting effectiveness in wheelchair basketball players across a competitive season: a preliminary study"

_PeerJ, doi:10.7717/peerj.15785_

## Round 0.1 · original submission · Minor Revisions

The paper is interesting, well-structured, and well-written. To improve it some minor suggestions should be followed.

Reviewer 1 ·

Basic reporting

No comment

Experimental design

No comment

Validity of the findings

No comment

Additional comments

General comment
Thank you for the opportunity to review this article. Congratulations, the paper is well written, the contents are presented in detail, as well as the results are clearly presented, and the tables are well done. It could provide relevant practical information.
The specific hypothesis is well formulated, and the rationale is consistent with current literature.
I have a few minor comments that can be addressed to improve the paper.
- Considering the first limitation of the study (i.e., that the investigations were conducted on a team), Please, I suggest adding the final wording in the title: “preliminary study”, “case study” or “team case study”.
- The materials and methods are very detailed and complete. I would only suggest, if it is possible, adding some more information regarding the members of the team considered.
- Your first aim indicates the need to identify "situational-related variables". I suggest improving the description of how the variables were identified in lines 233-236, to provide more clarity and completeness to the paper.
- I suggest increasing the graphic quality of Figure 1, so that it looks more attractive.

Minor points:
Line 66: Please, I recommend adding the “,” after “…in addition…”
line 101: the words Paralympic Games, should contain a capital letter "G" in "games”.
line 328: By "…relevant evel…" you meant “…relevant level…"?
-Finally, the English language could be improved to ensure better understanding and clearer form:
Line 79: Please I suggest "player also has…" rather than "player also have…"
Line 80: Please I suggest " Concerning the classification...." rather than "With reference to the classification..."
Line 90: Please, I suggest replacing "... integration the coaches’..." with "... integration of coaches’..."
Line 186: Please, I suggest replacing the verb "...interpretated..." with "...interpreted”.
Line 242: Please, I suggest replacing "...in relation with the number of shots..." with " ...in relation to the number of shots..."
Line 290-291: Please, I suggest substituting the phrase “This gives offensive players the opportunity to attempt their preferred shots early in a possession and, potentially....” to “This allows offensive players to attempt their preferred shots early in the possession and, potentially....”.
Line 302: Please, I suggest replacing "WB players reaches..." to "WB players reach..."

Reviewer 2 ·

Basic reporting

In the abstract, it is unclear where the conclusion begins.

In Line 156, the phrase "2 1/2 hours" is commonly used in casual or informal writing. I suggest using "2 hours and 30 minutes" or "150 minutes" for a more precise representation.

I am uncertain if the resolution has been enhanced prior to publication, but in Figure 1, the distinction between Zone 2 and Zone 3 is not clear.

Experimental design

no comment

Validity of the findings

This paper is highly interesting and well-structured. A major strength of this manuscript is its investigation of the field-goal effectiveness of non-elite high-point WB players throughout a competitive season, making it the first study of its kind. Additionally, the authors provide comprehensive support for their article through a comprehensive literature analysis.

---

## Round 0.2 · accepted · Accept

Authors have improved the paper following point by point the reviewers comments and suggestions.